# Understanding LLM Embeddings for Regression

**Eric Tang**                                                              *erictang000@gmail.com*
*Stanford University*
*Google DeepMind Academy Program*

**Bangding Yang**                                                          *jfyang@google.com*
*Google*

**Xingyou Song**                                                          *xingyousong@google.com*
*Google DeepMind*

**Reviewed on OpenReview:** *https:// openreview. net/ forum? id= Wt6Iz5XNIO*

## Abstract

With the rise of large language models (LLMs) for flexibly processing information as strings, a natural application is regression, specifically by preprocessing string representations into LLM embeddings as downstream features for metric prediction. In this paper, we provide one of the first comprehensive investigations into embedding-based regression and demonstrate that LLM embeddings as features can be better for high-dimensional regression tasks than using traditional feature engineering. This regression performance can be explained in part due to LLM embeddings over numeric data inherently preserving Lipschitz continuity over the feature space. Furthermore, we quantify the contribution of different model effects, most notably model size and language understanding, which we find surprisingly do not always improve regression performance.

## 1 Introduction and Related Work

*Regression* is a fundamental statistical tool used to model the relationship between a metric and a selected set of features, playing a crucial role in various fields, enabling predictions, forecasting, and the understanding of underlying relationships within data. Traditional regression techniques often rely on handcrafted features or domain-specific knowledge to represent input data. However, the advent of Large Language Models (LLMs) and their ability to instead process semantic representations of text has raised the question of whether regression can instead be performed over free-form text.

Previous works have predominantly examined the topic of LLM-based regression through *decoding*, i.e. generating floating point predictions using token-based sampling. For example, (Song et al., 2024) examines the case when the model is fully accessible and fine-tunable against data, while (Vacareanu et al., 2024) study the ability of *service-based* closed-source LLMs such as GPT-4 using in-context learning.

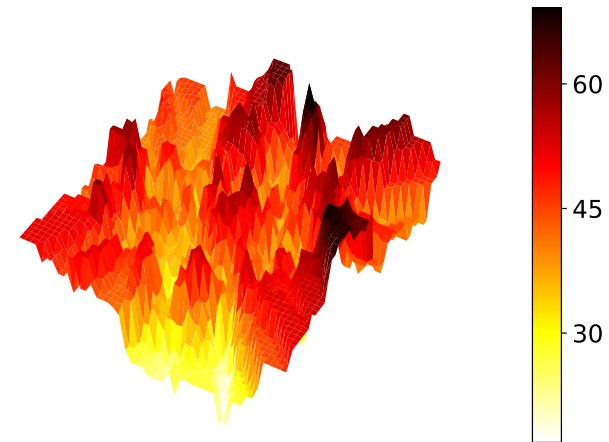

Figure 1: Rugged surface of a 5D Sphere function when inputs are represented as Gemini embeddings of dimension 6K+, post-processed by t-SNE into 2D space.

One understudied case however is the use of service-based LLM embeddings - fixed vector representations derived from pre-trained (but frozen) language models, which are ubiquitously offered among most LLM services (OpenAI, 2023; Google, 2024; Anthropic, 2024). Although they are used frequently in recent applications such as retrieval (Karpukhin et al., 2020), semantic similarity (Li et al., 2020), and a variety of other downstream language tasks (Liu et al., 2020), there has been very little fundamental research around their use in regression, outside of specific applications such as Bayesian Optimization (Nguyen et al., 2024; Kristiadi et al., 2024).

In contrast to decoding-based regression techniques, embedding-based regression allows the possibility of cheap data-driven training using inexpensive and customizable post-embedding layers such as multi-layer perceptrons (MLPs). However, as shown in Figure 1, when the domain of a simple function is expressed using high-dimensional embeddings, unexpected characteristics and irregularities can arise, prompting the need for a thorough analysis. Furthermore, LLMs by default are *not explicitly trained* for embedding-based regression, rather purely for token generation, and thus it is worth analyzing the emergent behaviors of LLM embeddings when applied to regression.

This paper investigates the behavior of these LLM embeddings when used as features for standard tabular regression tasks. Most notably, our findings are:

- LLM embeddings are *dimensionally robust*, i.e. regression performance can remain strong even over high-dimensional data, whereas traditional representations significantly suffer.
- Over numeric formats, LLM embeddings preserve Lipschitz-continuity and smoothness over feature space, which naturally enables regression when using a downstream MLP head.
- Factors which directly impact language understanding (e.g. size, pre-training, and input formatting) have more nuanced effects for regression and do not always provide significantly better outcomes.

## 2 Problem and Methodology

A regression task $\mathcal{T} = (f, \mathcal{X}, \mathcal{D})$ consists of an underlying scalar-valued function $f : \mathcal{X} \rightarrow \mathbb{R}$ over an input space $\mathcal{X}$. Provided are offline training data $\mathcal{D}_{train} = \{(x_1, y_1), ..., (x_T, y_T)\}$ collected from querying $f$ and an analogous test set $\mathcal{D}_{test}$ for evaluation. Given access to training data $\mathcal{D}_{train}$, the goal is to obtain accurate predictions over test points $(x, y) \in \mathcal{D}_{test}$, usually measured by an aggregate performance measure, e.g. mean squared error or Kendall-Tau ranking scores.

Required by nearly all learnable regression methods are *features*, which we assume come from an *embedder* $\phi : \mathcal{X} \rightarrow \mathbb{R}^d$ which takes an input $x$ and returns a fixed-dimensional feature representation, of dimension $d$. Here, we use the terms "features" and "embedding" interchangeably, since traditional methods typically use a canonical, manually defined feature engineering method for tabular data, in which continuous values are normalized and categorical selections are one-hot encoded. This feature vector $\phi(x)$ is then sent to a downstream predictor, e.g. MLP or random forest, which is trained using a loss function such as mean squared error.

Language models also provide a canonical definition of embedding, which typically consists of, in order:

1. Tokenizing a string representation $x$ into $L$ tokens.
2. Obtaining a "soft prompt" $\mathbb{R}^{L \times v}$ via vocabulary look-up.
3. Applying a forward pass of a Transformer to obtain an output $\mathbb{R}^{L \times f}$.
4. Pooling down to a fixed dimension vector in $\mathbb{R}^d$.

Afterwards, one may also attach an MLP predictor head and apply an analogous training procedure as in the traditional case. Thus we can see that the only difference becomes the input representation $\phi$, i.e. whether we used a traditional $\phi_{\text{trad}}$ or LLM-based $\phi_{\text{LLM}}$.

While it is straightforward to assume that the whole process outlined for LLMs should constitute the definition of a language model embedding $\phi_{\text{LLM}}$, it is not obvious how much each of these steps may contribute to the final regression result. For instance, one could simply skip applying a forward pass in step (3) and pool the soft prompt directly, or use a randomly initialized model as opposed to a pretrained one. We extensively study this case in Section 3.3.

### 2.1 Modeling Specifics

To minimize confounding factors and maintain fairness during comparisons, we use the exact same MLP prediction head (2 hidden layers, ReLU activation), loss (mean squared error), and $y$-normalization scheme (shifting by empirical mean and dividing by empirical deviation), regardless of using $\phi_{\text{LLM}}$ and $\phi_{\text{trad}}$. Note however, that the embedding dimensions of the two representations may be different, and so we distinguish them using notation $d_{\text{llm}}$ and $d_{\text{trad}}$ respectively, where typically $d_{\text{llm}} \gg d_{\text{trad}}$.

To demonstrate consistent results over different families of language models, we benchmark over both the T5 (Raffel et al., 2020) and Gemini 1.0 (Google, 2024) families, which use different architectures (encoder-decoder and decoder-only), different vocabulary sizes (32K and 256K), and embedding dimensions, respectively. Following (Li et al., 2020; Reimers & Gurevych, 2019), we use average-pooling as the canonical method of aggregating Transformer outputs, as empirical comparisons in Appendix A.4 found this was optimal. Thus the embedding dimension $d_{\text{llm}}$ is equivalent to the the output feature dimension $f$ following a forward pass, and specific sizes of $d_{\text{llm}}$ can be found in Appendix B.3.

Similar to previous work (Song et al., 2024; Nguyen et al., 2024), for string representations of $x$ from any regression task, by default we use a key-value JSON format with consistent ordering of keys, i.e. `{param1:value1,param2:value2,...}`, with specific examples shown in Appendix C.2.

### 2.2 Regression Tasks

Appendix C.1 contains full details on our regression tasks. We first use synthetic, closed-form functions in order to produce controlled studies in which we may query any $x$ from the input space. We use 23 functions defined from the standard Black-Box Optimization Benchmarking (BBOB) suite (Elhara et al., 2019), supporting continuous inputs of any dimension. To avoid confounding terminology between embedding "dimension" $d$ and the intrinsic "dimension" of an objective $f$, we denote the latter as "degree-of-freedom" (DOF), and thus $f(\cdot)$ is dependent on input coordinates $x^{(1)}, \ldots, x^{(\text{DOF})}$, each of which is between $[-5, 5]$. This provides a comprehensive variety of both convex and non-convex objective landscapes to regress upon.

We further use real-world regression tasks representative of those encountered in the wild and in industry settings by benchmarking over offline objective evaluations over production systems, collected from hyperparameter tuning records. These consist of four families, with each family containing at least 50 individual yet similar regression tasks with varying amounts of data. The families are:

- AutoML (Google Cloud, 2023): Automated Machine Learning service for Tensorflow Extended (Google, 2023) pipelines (e.g. batch size, activation, layer counts) over tabular or text data.
- Init2Winit (Dahl et al., 2023): Learning rate scheduling parameters influencing common image classification tasks (e.g. ResNets on CIFAR-10 and ImageNet).
- XLA (Phothilimthana et al., 2021): Tuning for the Accelerated Linear Algebra (XLA) compiler which affects LLM serving latencies.
- L2DA (Yazdanbakhsh et al., 2021): "Learning to Design Accelerators", for improving accelerators such as TPUs and corresponding computer architectures to improve hardware performance.

In the real world regression tasks, each parameter may be continuous or categorical, and we define the DOF of such a task by its number of parameters. Note that for synthetic objectives, where all inputs are continuous, $d_{\text{trad}} = \text{DOF}$. However, for real-world tasks with categorical parameters, $d_{\text{trad}} > \text{DOF}$ due to additional one-hot encodings.

For obtaining data, we may either sample $(x, y)$ pairs (in the case of synthetic objectives where $x$ are uniformly sampled from $\mathcal{X}$), or use the given offline data (in the case of real-world tasks, where they were actual evaluations from an optimization trajectory), using a standard 8-1-1 train-validation-test split.

Due to the inherent differing of metric scales across tasks, it would be inappropriate to aggregate results based on scale-dependent metrics such as mean squared error (MSE). Furthermore, we found that the selection of the regression metric (e.g. Kendall-Tau, Pearson, mean squared error, mean absolute error) did not matter for comparisons, as they all strongly correlated with each other. Thus, by default we report the Kendall-Tau ranking correlation, which is always within $[0, 1]$ and can also be aggregated across different tasks.

## 3 Experimental Results

### 3.1 High Dimensional Regression

We begin by demonstrating cases in which LLM embeddings better represent inputs over high degree-of-freedom spaces than traditional representations. In Figure 2, we show that for a subset of functions, LLM embeddings possess surprising robustness, retaining the same performance for varying DOFs whereas traditional baselines such as XGBoost and MLPs significantly falter over higher DOFs. Appendix A.5 demonstrates this is inherent to the LLM embeddings, regardless of the regression head used (e.g., XGBoost).

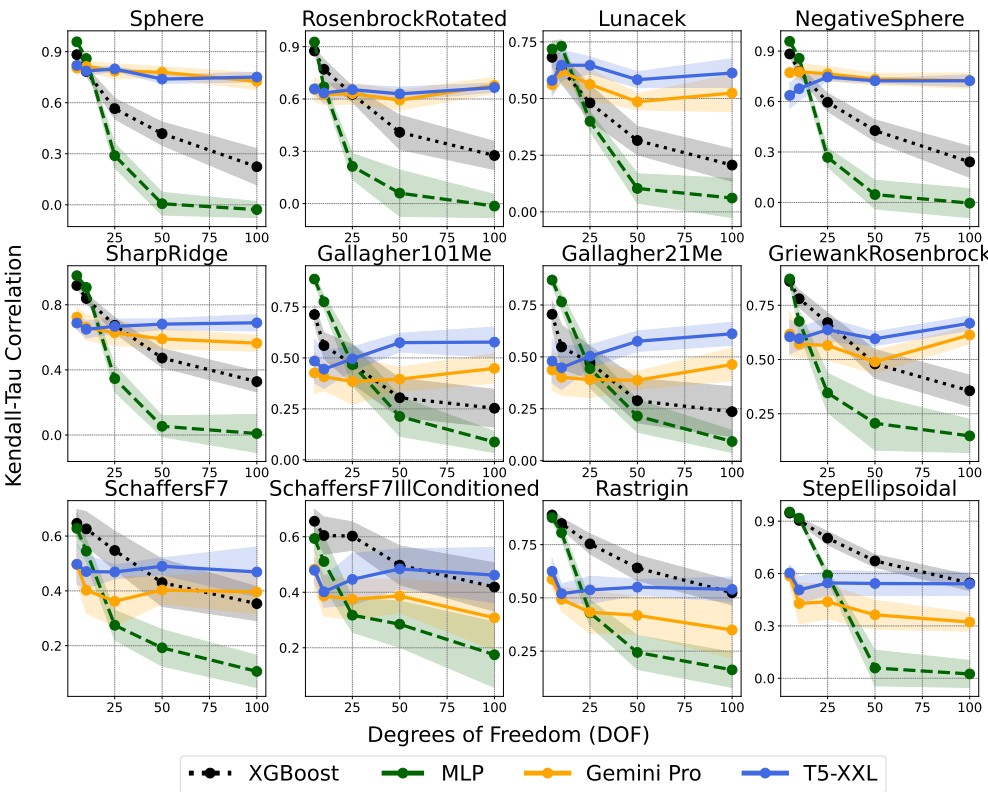

Figure 2: Higher (↑) is better. Degrees of freedom (DOF) vs Kendall-Tau correlation for various BBOB functions. Results are averaged over 12 runs of random data generation. Each task's data consists of 500 $(x, y)$ evaluations sampled uniformly across the input space, using a 8-1-1 split for train-validation-test.

This result is not universal however, as we show in Appendix A.1, this pattern does not apply for a few selected functions, but nonetheless it occurs in the majority of the BBOB functions. We further corroborate this observation over real-world tasks in Table 1. We see that in general, regressions on LLM embeddings outperform traditional methods more often for tasks with higher DOFs (AutoML and XLA).

| Task Name | Avg. DOF | T5-Small % | T5-XXL % | Gemini Nano % | Gemini Pro % |
|---|---|---|---|---|---|
| Init2Winit | 4 | 6.7 | 8.0 | 11.3 | 19.0 |
| L2DA | 10 | 2.7 | 12.0 | 9.3 | 10.7 |
| AutoML | 29 | **30.7** | **41.3** | **29.3** | **36.0** |
| XLA | 35 | 17.2 | 29.3 | 18.9 | 24.1 |

Table 1: Percentage of tasks in which $\phi_{\text{LLM}}$ outperforms $\phi_{\text{trad}}$ across various real world regression tasks. Results reported for 75 tasks per family, except for XLA, which only contains 58 tasks. Full results in Appendix A.2.

### 3.2 LLM Embedding Smoothness

Particularly due to the discrete nature of tokenization, it is non-obvious whether LLM embeddings possess a notion of continuity in embedding space. For example, assuming character-wise tokenization, `1.234` is not so numerically distant from `1.567`, but is *token-wise* distant, as the majority of the tokens (`234` and `567`) are not shared. The notion of continuity and smoothness is crucial for neural network generalization (Kalimeris et al., 2019; Neyshabur et al., 2018), robustness (Weng et al., 2018), vulnerability to adversarial examples (Goodfellow et al., 2015), and more. We can characterize smoothness in the regression case by the *Lipschitz-continuity* induced by a representation $\phi$ in its latent space $\mathbb{R}^d$.

Intuitively, similar inputs should lead to similar objective values, which can be quantified inversely by the Lipschitz factor $L(x, x') = \|f(x) - f(x')\| / \|\phi(x) - \phi(x')\|$ with respect to a representation $\phi$ and $\|\cdot\|$ norm. We emphasize to the reader that the input space $\mathcal{X}$ *does not actually have an explicit notion of distance on its own*. Instead, traditionally it has always been *assumed* that the distance was defined canonically by Euclidean distance over the traditional embedding method, i.e. $\|\phi_{\text{trad}}(x) - \phi_{\text{trad}}(x')\|_2$ as demonstrated by common use of Euclidean-based radial basis and Matern kernels (Genton, 2002) during regression modeling. However, as seen from the results previously, it may be the case that $\phi_{\text{trad}}$ is suboptimal for some regression tasks.

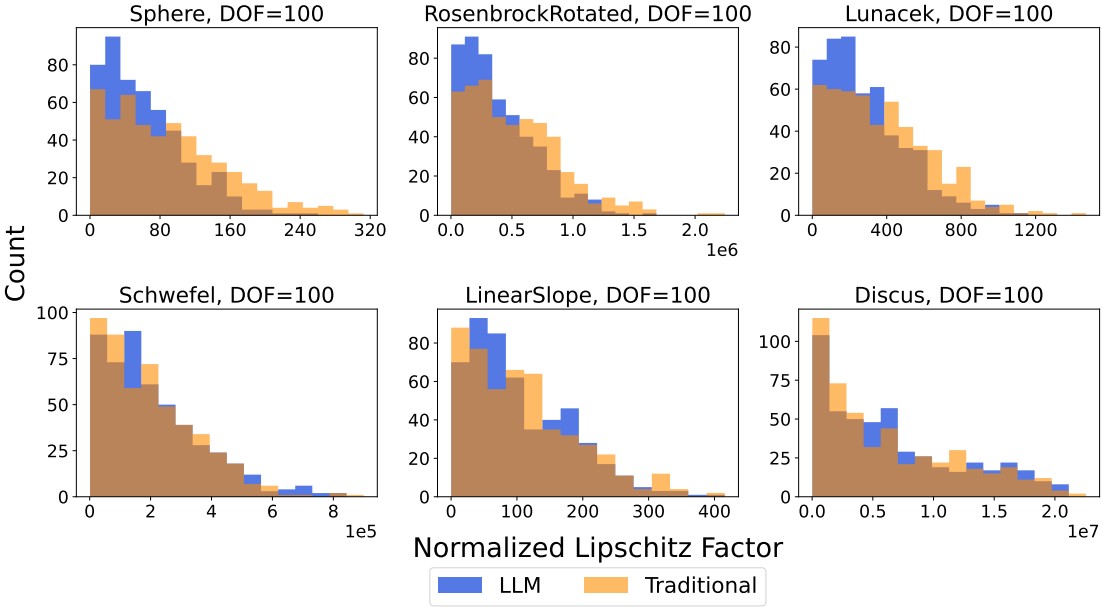

Figure 3: Left-skewness ($\leftarrow$) is better. NLFDs induced by $\phi_{\text{LLM}}$ (T5-XXL) and $\phi_{\text{trad}}$. **Top:** Cases where $\phi_{\text{LLM}}$ outperforms $\phi_{\text{trad}}$ for regression. **Bottom:** Vice-versa where $\phi_{\text{trad}}$ outperforms $\phi_{\text{LLM}}$.

In order to analyze the continuity of an embedding $\phi$ with respect to offline data $\mathcal{D}$, we define a Normalized Lipschitz Factor Distribution (NLFD) as follows:

1. Full-batch normalize, i.e. apply shifting and scaling to each $\phi(x)$ so that in aggregate, $\mathcal{D}$ has zero mean and unit variance per coordinate.
2. For each $x \in \mathcal{D}$, choose $x' \in \mathcal{D}$ such that $\phi(x')$ is the nearest $\ell_2$ neighbor of $\phi(x)$, and compute the Lipschitz factor $L(x, x')$.
3. To assume an average embedding norm of 1 for different embedding dimensions $d$, we downscale all Lipschitz factors by $\sqrt{d}$.

We see that there is a high inverse relationship between the skewedness of the NLFD and regression performance. Specifically, in Figure 3, when $\phi_{\text{LLM}}$ outperforms $\phi_{\text{trad}}$ for regression, $\phi_{\text{LLM}}$'s distribution of Lipschitz factors also tends to skew relatively more to zero than $\phi_{\text{trad}}$, and vice-versa.

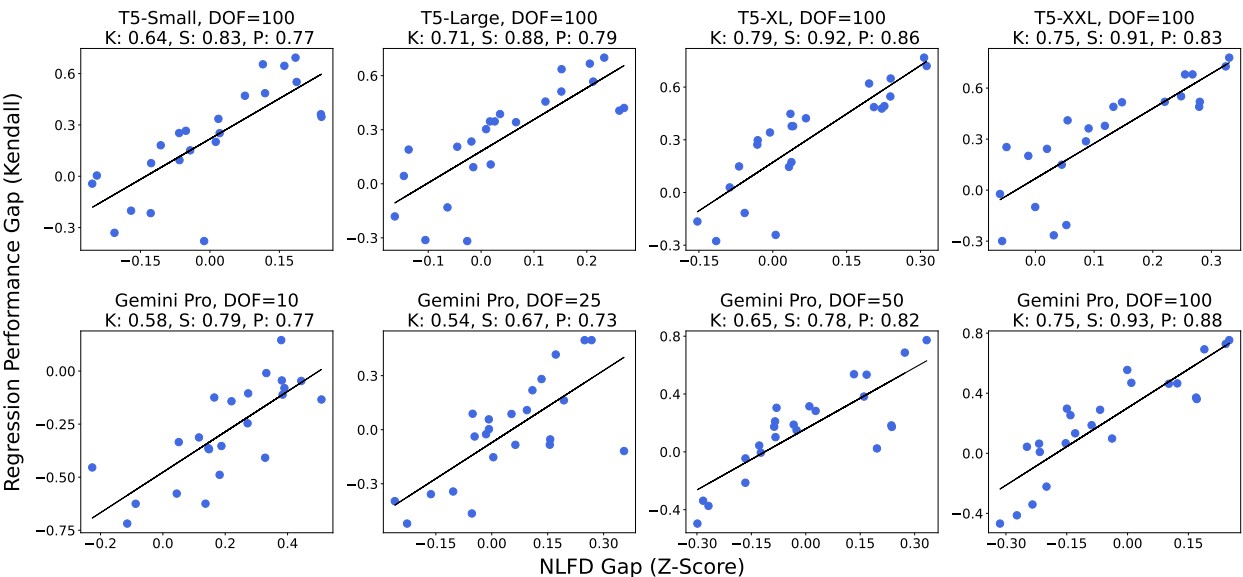

Figure 4: Relationship between gaps in NLFD (via Z-score) and regression performance for all 23 BBOB functions. Relationship is quantified using (K, S, P), which respectively are Kendall-Tau, Spearman and Pearson correlations. **Top:** We vary model size within the T5 model family. **Bottom:** We vary the objective's DOF for Gemini Pro.

To formally quantify comparisons between NLFDs from $\phi_{\text{LLM}}$ and $\phi_{\text{trad}}$, for a fixed regression task, we may thus compute the Z-score using the difference of the two distributions:

$$Z = \frac{\mu_{\phi_{\text{trad}}} - \mu_{\phi_{\text{LLM}}}}{\sqrt{\sigma^2_{\phi_{\text{trad}}} + \sigma^2_{\phi_{\text{LLM}}}}} \tag{1}$$

where $\mu_\phi$ and $\sigma_\phi$ are respectively mean and standard deviations of the NLFD of a representation $\phi$. We may then observe the relationship between gaps in representation smoothness vs. regression performance. In Figure 4 with extended results in Appendix A.3, we see that for a given BBOB regression task, the Z-score (i.e. gap in embedding smoothness) is highly correlated with the gap in regression performance, regardless of the model used (T5 or Gemini) or the DOF of the underlying objective $f$.

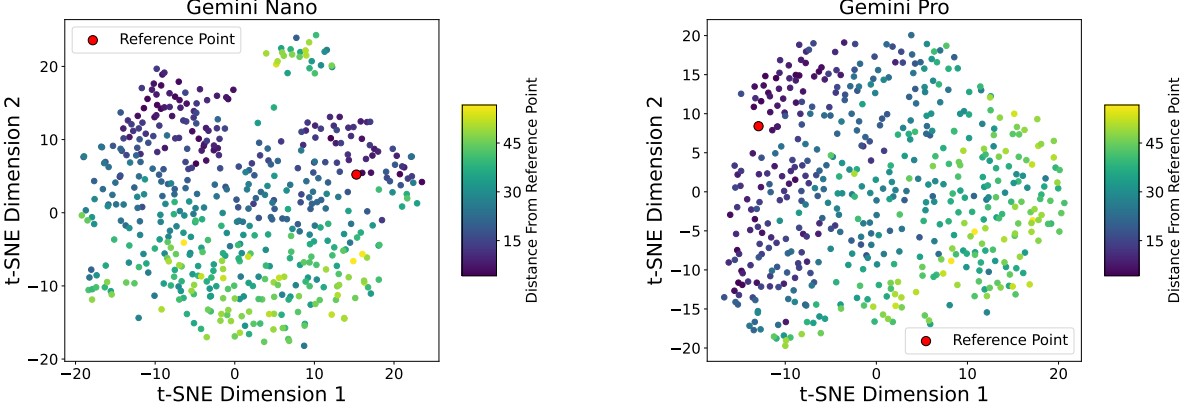

Figure 5: t-SNE for Gemini (Nano and Pro) embeddings of points sampled around a DOF=100 reference point. Traditional $\ell_2$ distance is overlaid in color.

In Figure 5, we further visualize whether $\phi_{\text{LLM}}$ is distance aware, i.e. whether $\phi_{\text{LLM}}(x)$ are $\phi_{\text{LLM}}(x')$ are close in embedding space if $\phi_{\text{trad}}(x)$ and $\phi_{\text{trad}}(x')$ are close. As mentioned before however, there is no ground truth notion of "closeness" - nonetheless, we use $\phi_{\text{trad}}$ as a point of comparison. Since it is inappropriate to

simply sample $x$'s uniformly in a high DOF space, as then average distances concentrate around $\sqrt{\text{DOF}}$, we instead take a reference point and sample points from $\ell_2$-balls of increasing distance from the reference. We see that distances over the LLM embedding space are correlated with the traditional measure of distance, but may be non-linearly warped, which benefits LLM-based regression in certain cases as seen in Section 3.1.

## 3.3 Model Effects

In this subsection, we comprehensively investigate the impact of many common LLM factors such as model size and pretraining on regression performance. Additionally, we also ablate the effects of different string representations and LLM subcomponents such as vocabulary embeddings.

**Are Larger Models Always Better?** Within the research community, the prevailing assumption is that there exists a direct correlation between language model size and performance improvement. However, with the rise of leaderboards such as LMSYS (LMS, 2023), smaller models have been shown to outperform larger competitors, due to differences in their "recipe", such as training data quality, pre-training and post-training techniques, and architecture.

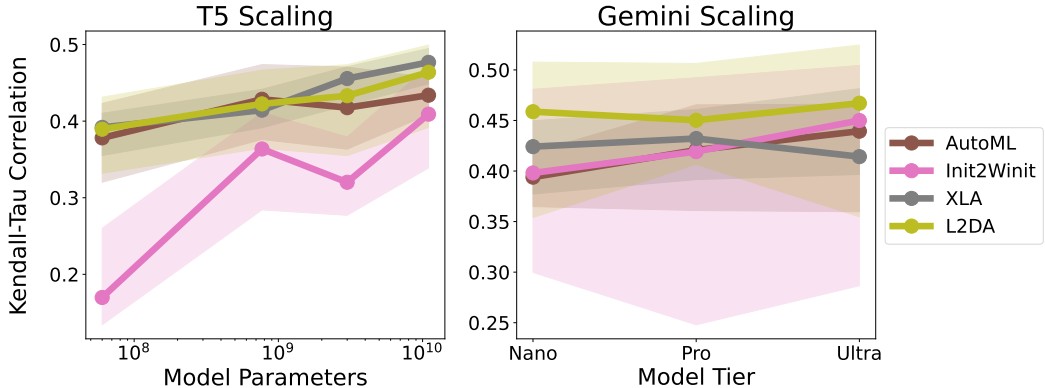

Figure 6: Higher (↑) is better. Model size vs regression performance on hyperparameter tuning tasks across T5 and Gemini model families. Median performance is plotted, along with 40-60 percentiles as error bars.

In Figure 6, we see that over various real world regression tasks, T5 models exhibit a clear trend of improved performance when increasing model size, when training methodology is fixed. In contrast, model tiers within the Gemini family exhibit substantial variance, and larger model sizes do not consistently translate to superior results. We hypothesize this is due to differences in Gemini "recipes", as e.g. different model tiers may have used different pre-training datasets, architecture tweaks, and post-training configurations, whereas all T5 model sizes have only been pre-trained on the C4 web crawl corpus.

**Does Language Understanding Actually Help?** Recent works (Li et al., 2020; Devlin et al., 2019) have claimed that logit-based embeddings mostly measure the semantic similarity between string inputs, and thus it is unconfirmed whether they may be beneficial for numeric regression tasks. To resolve this, using the T5 family, we compare against using (1) a randomly initialized model for the forward pass, and (2) representing our features via vocabulary embeddings without applying a forward pass.

We first perform our study over BBOB tasks which use purely numeric tokens, where one may question the benefit of language pretraining over English corpus data. Surprisingly, Figure 7 shows that applying forward passes of pretrained models still helps, especially with larger sizes. However, it is worth mentioning that even features from randomly initialized models and vocabulary embeddings alone *are also dimensionally robust*, as they do not suffer performance drops with higher DOFs. For real-world tasks in Figure 8, it is more expected that applying forward passes by larger pretrained models helps. However, there is more variance in the improvement gaps, as the improvement is surprisingly quite minimal for some tasks such as AutoML and L2DA, while significant for Init2Winit and XLA.

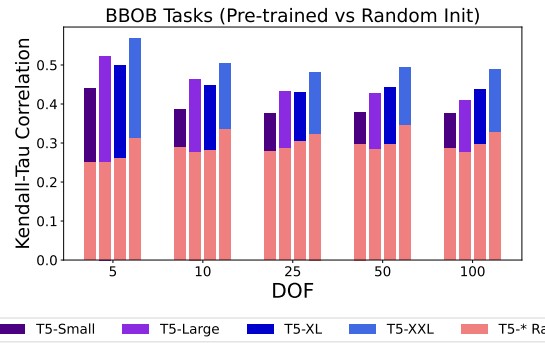 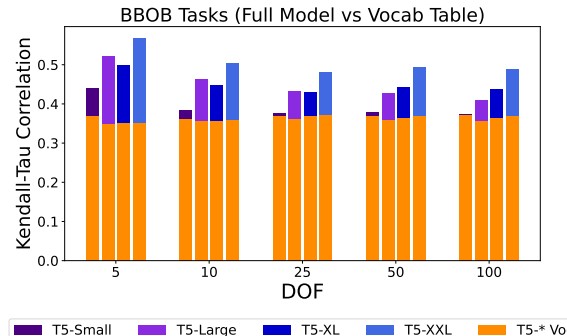

Figure 7: Kendall-Tau regression comparisons when comparing to random initialization (left) and vocabulary embeddings (right) on BBOB regression tasks. Each bar is averaged across all BBOB functions for a given model size and DOF.

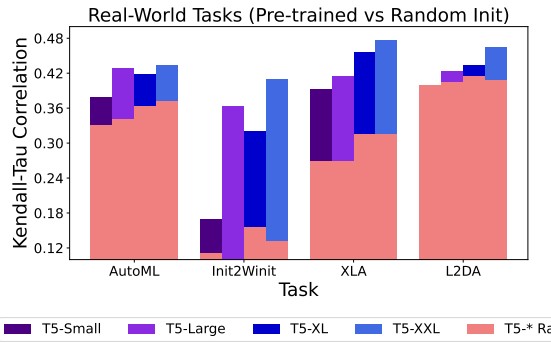 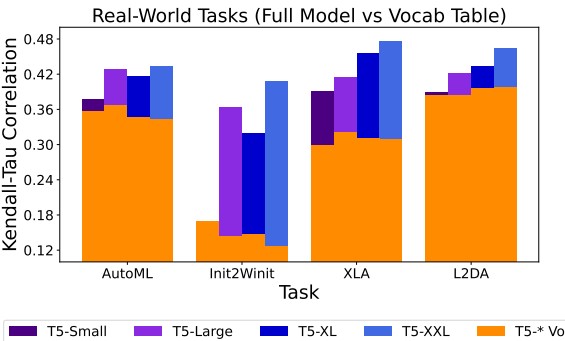

Figure 8: Same as Figure 7, but applied for real world regression tasks. Each bar is averaged across 75 tasks per family.

We further ablate differences in string representation, i.e. whether by default to show feature names as {param1:value1, param2:value2,...} or omit them, only showing [value1,value2,...]. In Figure 9, for the majority of tasks, omitting feature names does not significantly affect performance, although specific tasks such as XLA do benefit from feature names. This is surprising, as presumably feature names in XLA tasks such as auto_cross_replica_sharding are not as common as names such as batch_size or learning_rate found in both AutoML and Init2winit.

The results of Figures 8 and 9 combined lead to additionally surprising conclusions, such as language-to-numeric transfer. For instance, inputs $x$ from Init2Winit tasks only possess numeric values, and as expected, removing feature names does not significantly change regression results. Yet applying forward passes by pre-trained T5 models still benefits regression, despite the fact that T5's pre-training data contains mostly web-corpus data which is unlikely to contain significant amounts of scientific or numeric information (Dodge et al., 2021).

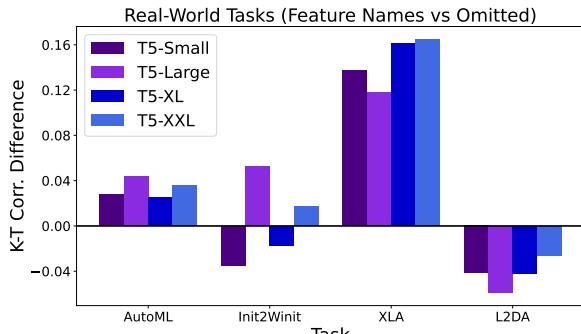

Figure 9: Difference in Kendall correlation when using full dictionary containing feature names, or only values.

**More Training Data Reduces Baseline Gaps:** Intuitively, as more samples are available in a task, the difference in inductive biases between regression methods should matter less, since predictions will be more influenced by training data. We verify this in Figure 10, where we see that for tasks with low numbers of

$(x, y)$ points, there is more variability in performance between using $\phi_{\text{LLM}}$ and $\phi_{\text{trad}}$, but additional training points decreases these differences.

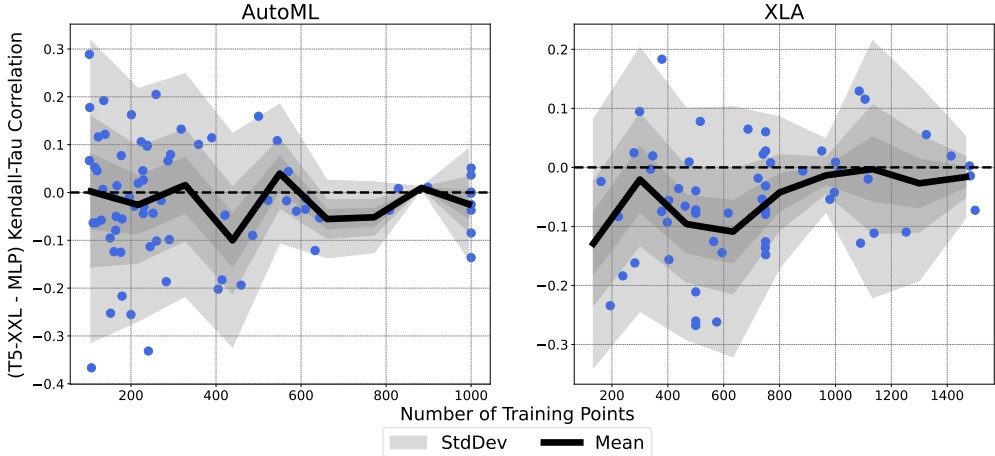

Figure 10: Performance gap between an MLP baseline and regression over T5-XXL embeddings for individual trials within the AutoML and XLA task settings. Higher ($\uparrow$) is better for LLM embeddings. Error bars are plotted for $\{0.5, 1.0, 2.0\}$ of the standard deviation.

## 4    Discussion: Limitations and Extensions

In this work, our emphasis was to provide more in-depth understanding of LLM embeddings with respect to regression. While we found many different cases in which they outperform traditional representations, we cannot broadly claim that LLM embeddings should always be used in serious applications of regression. Below, we list some limitations of our work and more potential areas of exploration.

**Different Modalities and Inputs:** Further investigation is needed to understand how LLM embeddings perform with non-tabular data, including combinatorial objects such as graphs, trees, and other complex structures, but also diverse modalities such as images, videos, and audio data.

**Prompt Formatting:** While we investigated the effects of parameter names for tabular string representations, we did not investigate the effects of different numeric representations. Since LLMs are predominantly pre-trained over human-written text, our $x$ formats also follows, e.g. 1234.5 is serialized directly into `1234.5`. However, these numbers may also be represented using scientific notation (e.g. `1.23e3`) or even customized variants, e.g. `[1 10e2 2 10e1 3 10e0 4 10e-1 ]` as in (Nogueira et al., 2021). We suspect that the fundamental conclusions would remain similar, although the specific numeric results may change.

**Different LLM Services:** Our work focuses on "depth" of understanding rather than "breadth" of results, although we did find many similar conclusions for both the T5 and Gemini model families, which are quite different in architecture, pre-training data, and post-training. While it remains to be empirically verified, we hypothesize similar conclusions may occur with other model families, such as GPT-4 (OpenAI, 2023), Claude (Anthropic, 2024), and LLaMA (Touvron et al., 2023), especially as they share fundamentally similar approaches with Gemini.

**Different Embedding Definitions:** As mentioned in the main body, using average-pooling for LLM "embeddings" follows previous literature for consistency, is one of the simplest methods to obtain a fixed dimensional feature vector. and is also better than using max-pooling or last-token logits as shown in Appendix A.4. However, other definitions have been proposed, such as collecting intermediate outputs (Chen et al., 2022). Such additional methods are worth studying in the future for understanding their effects on regression.

**In-Context Learning (ICL):** An alternative prompting method, particularly natural for decoder-based architectures, would be to place all previous evaluations in the context as "shots" and obtain only the logits

from a query $x$. However, this can severely limit the amount of training data allowed, as the context window still has a finite maximum length. In this paper, we primarily focused on the zero-shot case where the context window only contains the query, which allows the downstream MLP to train over unbounded amounts of data.

**Computational Costs:** Compared to traditional regression techniques which can even be run on CPUs, LLM inference almost always requires accelerator usage, making them more expensive if needed in serious regression tasks. Remote procedure calls to service-based LLMs such as Gemini also adds an additional layer of latency. However, compute costs for inference are orders of magnitude cheaper than for training, and typically only require a few GPUs or TPUs, making embedding-based regression still very feasible for most academic labs or industries.

**Smoothness Computations:** One limitation is that our smoothness analysis is only feasible when one has online access to $f(\cdot)$ as in the case of BBOB functions but not offline real world data, since one needs to obtain arbitrarily close $(x, x')$ pairs to understand local behaviors within the feature space. Our analysis however, may be extendable to any space $\mathcal{X}$ (e.g. combinatorial) which admits a distance metric.

## 5 Conclusion

We thoroughly investigated multiple important aspects around the use of LLM embeddings for traditional regression. We found that LLM embeddings can be quite performant for input spaces with high degrees of freedom, and proposed the Lipschitz factor distribution to understand the embedding-to-objective landscape and its relationship to regression performance. We further investigated the nuanced conditions for which better language understanding does improve LLM-based regression.

## Acknowledgments

We thank Yutian Chen, Daniel Golovin, Chansoo Lee, Tung Nguyen, and Sagi Perel for relevant discussions during experimentation and the writing of this paper. We further thank the organizers of the Google DeepMind Academy Program for providing the opportunity to do this research.

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

# Appendix

## A  Extended Experiments

### A.1  High Dimensional Regression

For full transparency, In Figure 11, we display BBOB functions where LLM-based regression was not consistently dimensionally robust against MLP and XGBoost baselines. Note that even in these cases, we still see certain cases where a language model outperforms at least one of the baselines, e.g. in the Discus and DifferentPowers functions, Gemini and T5 outperform MLP but not XGBoost.

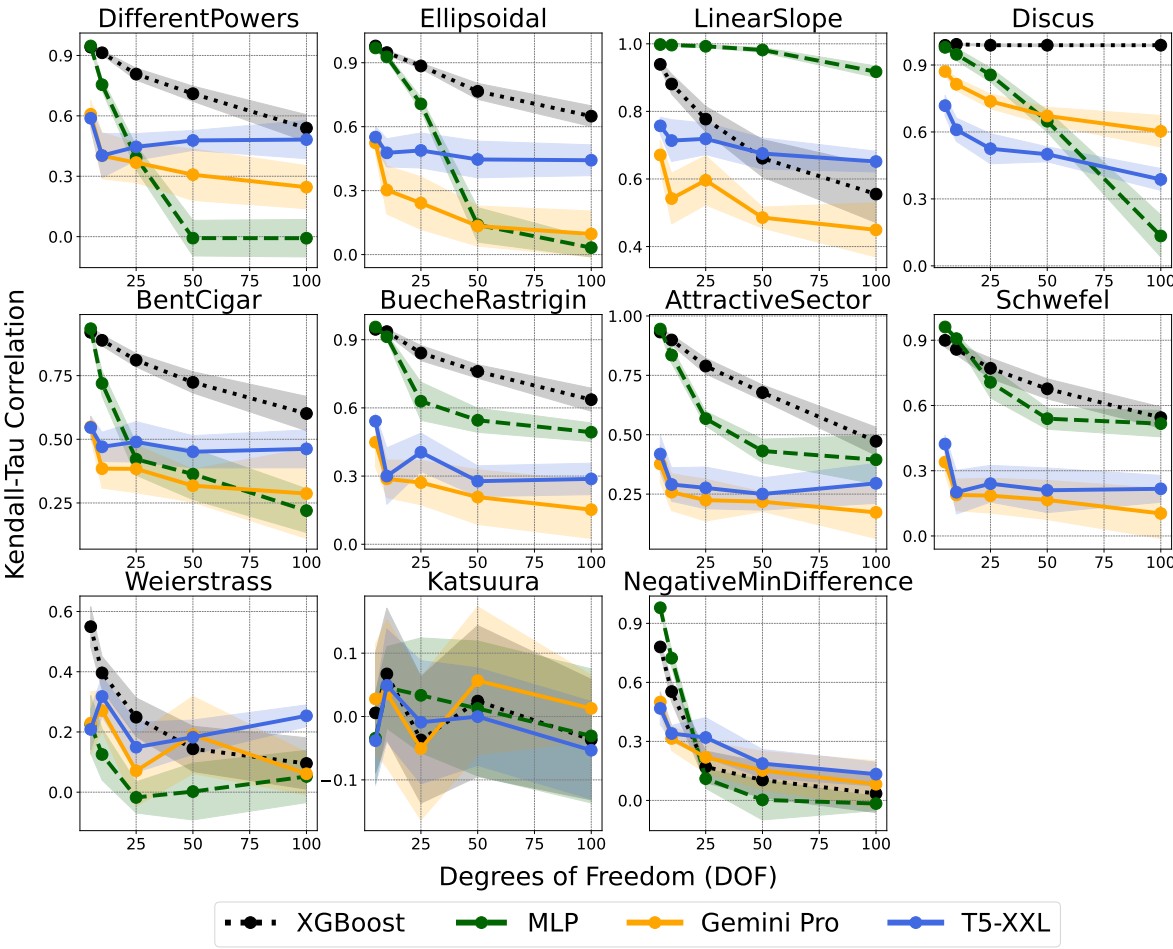

Figure 11: Following Figure 2 in the main body, we present BBOB functions in which LLM embeddings did not completely outperform traditional baselines.

## A.2    Real World Results

Despite Table 1 of the main body showing that there were numerous cases where LLM embeddings outperform traditional ones, we remind the reader in Figure 12 that *on average*, LLM embeddings still slightly underperform.

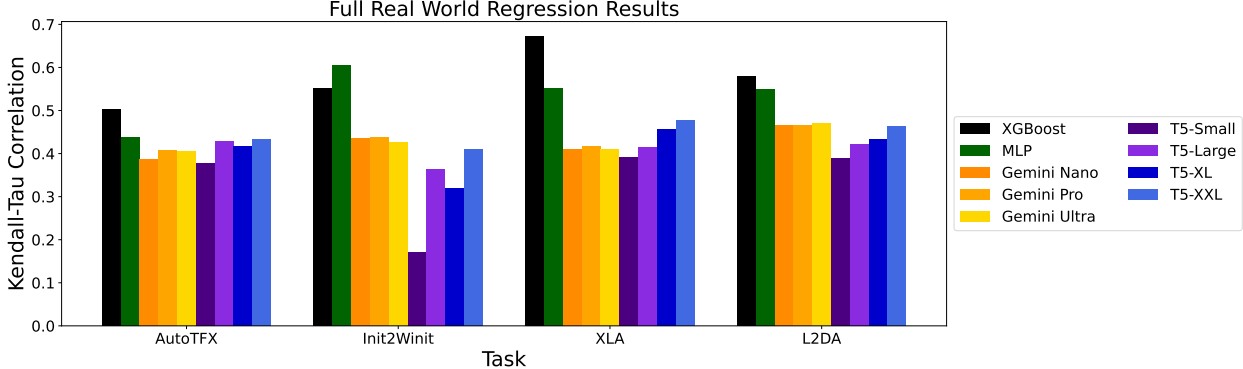

Figure 12: Full Results over real world tasks. Displayed is the mean Kendall-Tau Correlation over all tasks within each family.

## A.3    Performance Correlations

Following Figure 4, in Table 2, we see that the relationship between the smoothness induced by the embedding and the performance in regression is consistent throughout.

| Model | DOF=5 | DOF=10 | DOF=25 | DOF=50 | DOF=100 |
|---|---|---|---|---|---|
| Gemini Nano | 0.81 | 0.81 | 0.70 | 0.75 | 0.86 |
| Gemini Pro | 0.78 | 0.77 | 0.72 | 0.82 | 0.88 |
| T5-Small | 0.75 | 0.76 | 0.79 | 0.79 | 0.76 |
| T5-Large | 0.78 | 0.73 | 0.79 | 0.85 | 0.79 |
| T5-XL | 0.82 | 0.60 | 0.80 | 0.86 | 0.85 |
| T5-XXL | 0.72 | 0.76 | 0.82 | 0.83 | 0.83 |

Table 2: Full set of data for Pearson correlation $\rho$ between Kendall's regression performance and gap in NLFD between input and embedding space for regression on all 23 BBOB functions, over DOF=$[5, 10, 25, 50, 100]$.

## A.4 Pooling Ablations

We ablate the effects of using different pooling mechanisms on T5-XXL logits, and see that average-pooling is consistently the best among the three standard pooling methods (details in Appendix B.2). We further see that using last-token logits not only performs the worst in absolute performance, but also is generally *not* dimensionally robust in comparison to average-pooling or max-pooling.

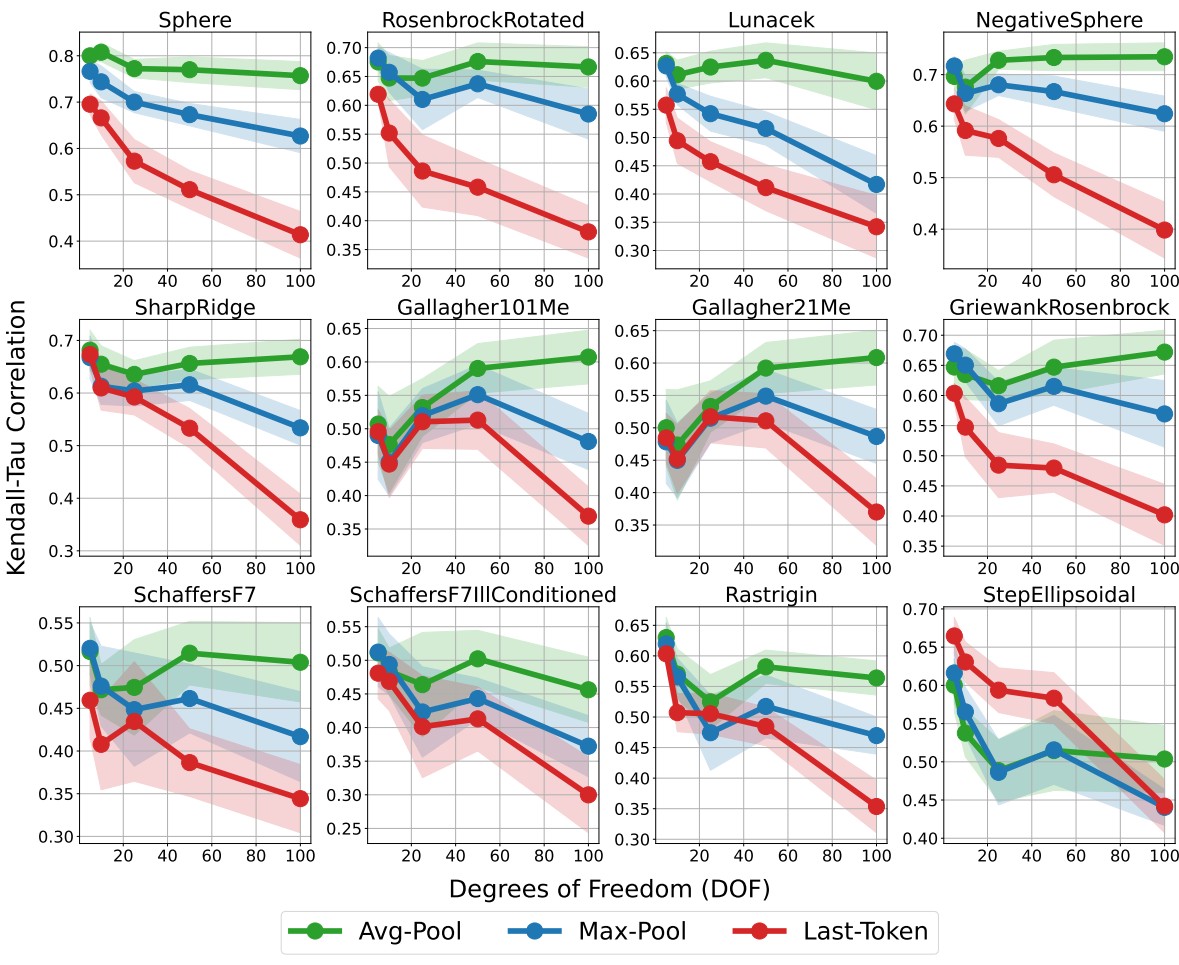

Figure 13: Higher (↑) is better. Follows the same setting as Figure 2, but comparing different embedding pooling methods.

## A.5 XGBoost for LLM Embeddings

In Figure 2 of the main body, XGBoost outperforms MLP when using $\phi_{\text{trad}}$ as input features. However, this pattern does not hold when using $\phi_{\text{LLM}}$, as shown in Figure 14. Here, XGBoost consistently underperforms against an MLP as a regression head for T5 embeddings across all BBOB functions, model sizes, and DOFs.

Despite this difference in head performance, the key observation remains: T5 embeddings exhibit dimensionality robustness and improve with model size regardless of whether XGBoost or an MLP is used as the regression head. This finding suggests that the quality of LLM embeddings is a fundamental property, independent of the specific regression technique employed.

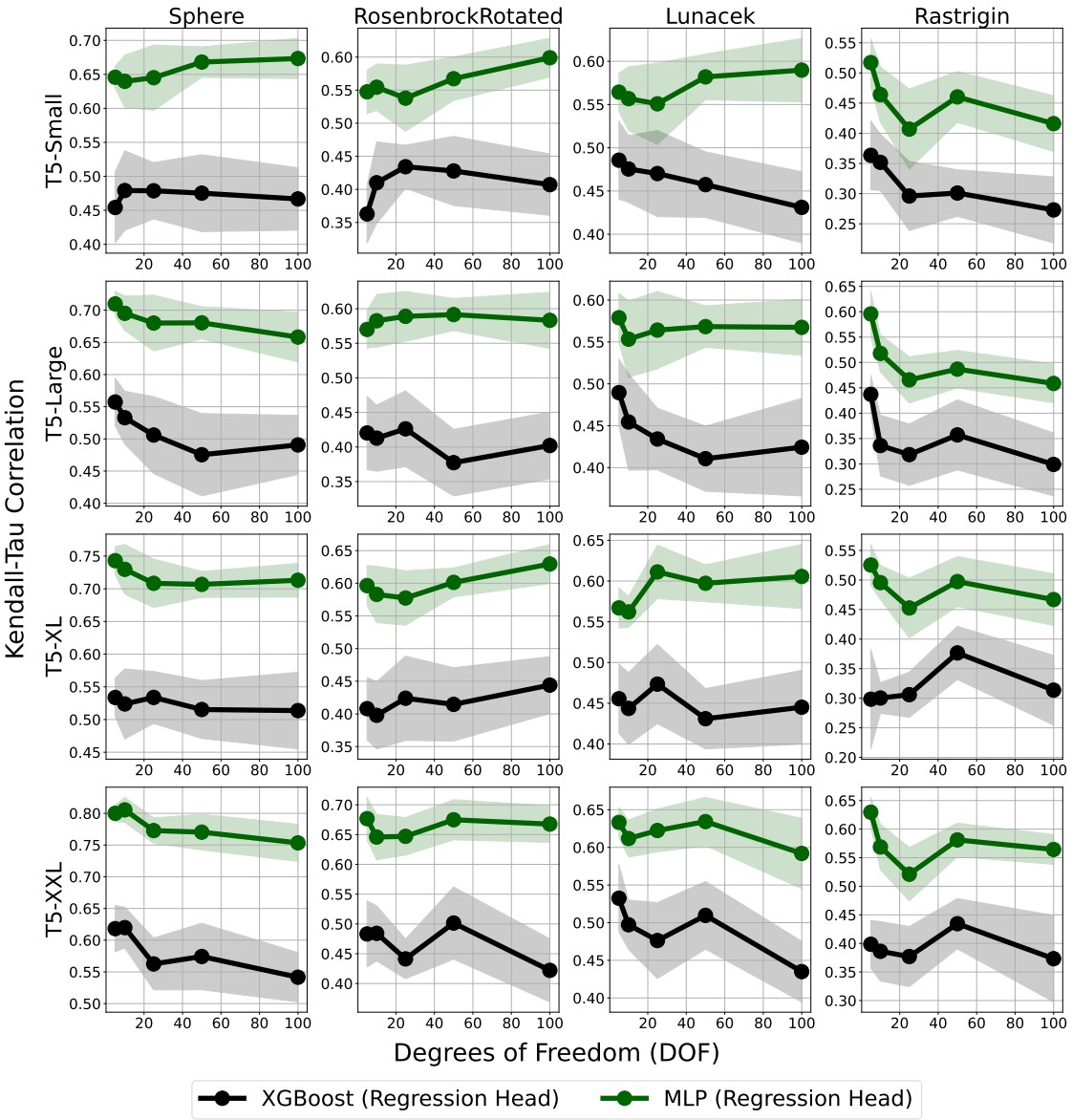

Figure 14: Higher (↑) is better. Follows the same setting as Figure 2 but comparing XGBoost and MLP over $\phi_{\text{LLM}}$, when varying BBOB functions (horizontal axis) and T5 model sizes (vertical axis).

# B    Exact Modeling Details

## B.1    Hyperparameters Used

The full list of hyperparameters and training details for MLP-based regression (using traditional and language model features):

- Regression Head: MLP with 2 ReLU hidden layers of dimension 256.
- Input-Normalization: We linearly scale each coordinate in $\phi$ to $[-1, 1]$, using its minimum and maximum observed values as the original endpoints.
- $y$-Normalization: We compute the empirical mean $\mu$ and standard deviation $\sigma$ over all $y$-values in the task's training data, and apply $y \leftarrow (y - \mu)/\sigma$ as a preprocessing step.
- Optimizer: AdamW with sweeped learning rates across {1e-4, 5e-4, 1e-3, 5e-3, 1e-2} and weight decay across {0, 1e-1, 1}.
- Loss: Mean Squared Error.
- Maximum Epochs: 300, with early stopping enabled.

Our XGBoost uses the same input normalization method as the MLP. We additionally grid-searched over the following parameters for each task:

- `"min_child_weight"`: $[1, 5, 10]$
- `"learning_rate"`: $[0.001, 0.01, 0.1]$
- `"gamma"`: $[0.0, 0.3, 0.5]$
- `"subsample"`: $[0.6, 0.8, 1.0]$
- `"colsample_bytree"`: $[0.6, 0.8, 1.0]$
- `"max_depth"`: $[3, 5, 7]$

## B.2    Pooling Mechanisms

Tokenized strings may be appended with additional tokens to the maximum sequence length $L$. Thus for example, before a forward pass, a token sequence may actually appear as $[t_1, t_2, \ldots, t_k, 1, 0, 0, 0, \ldots]$ where $t_i > 1$ and 0 and 1 are padding and optional end-of-sequence (EOS) tokens, respectively. Below, we specify how each pooling mechanism performs postprocessing steps over logits.

- Average-Pooling (default): Zero-out padding token logits, and average across the entire length axis.
- Max-Pooling: Same as Average-Pooling, but take the maximum across the length-axis for each coordinate.
- Last-Token: Only obtain the logit of $t_k$ or the EOS token, similar to the `<CLS>` token method but applicable to decoder-only methods as well.

Note that while Average-Pooling will have its coordinates scaled down by the maximum length $L$, input normalization from the regression head will re-scale them back to valid ranges.

## B.3    Embedding Sizes

Table 3 displays the embedding $d_{\text{llm}}$ for each model used in our experiments. As mentioned in the main text, note that $d_{\text{llm}}$ is significantly larger than $d_{\text{trad}}$.

| T5 Model | $d_{\text{llm}}$ |
|---|---|
| Small | 512 |
| Large | 1024 |
| XL | 2048 |
| XXL | 4096 |

| Gemini Model | $d_{\text{llm}}$ |
|---|---|
| Nano | 1536 |
| Pro | 6144 |
| Ultra | 14336 |

Table 3: Embedding dimensions $d_{\text{llm}}$ for T5 and Gemini model families.

## C   Extended Data Details

### C.1   Additional Data Details

**BBOB:** The Black-Box Optimization Benchmarking (BBOB) suite (Elhara et al., 2019) consists of 23 synthetic objectives which supports evaluation over continuous inputs of any DOF. Such functions comprehensively cover different landscape conditions (e.g. separability and optimality). Every function $f(x)$ by default has a global minimum $f(x) = 0$ at exactly the zero-vector input. Example of such functions include:

- Sphere: $f(x) = \sum_{i=1}^{DOF} (x^{(i)})^2$
- Rastrigin: $f(x) = 10 \left( DOF - \sum_{i=1}^{DOF} \cos(2\pi x^{(i)}) \right) + \|x\|^2$
- BentCigar: $f(x) = (x^{(1)})^2 + 10^6 \sum_{i=2}^{DOF} (x^{(i)})^2$

**Real World Data:** We use an extensive set of hyperparameter and blackbox optimization objectives, as these also tend to possess varied landscapes suitable for regression benchmarking.

- **AutoML:** Objectives collected from the Vertex AI platform for automated ML model selection and training for tabular or text data. For tabular data, Vertex AI searches over a tree of model and optimizer types, their hyperparameters, data transformation, and other components in the ML pipeline. For text, Vertex AI trains an ML model to classify text data, extract information, or understand the sentiment of the authors. For more information, see: `https://cloud.google.com/vertex-ai?#train-models-with-minimal-ml-expertise`.
- **Init2Winit:** Data from running deterministic, scalable, and well-documented deep learning experiments, with a particular emphasis on optimization and tuning experiments (e.g. ResNets on CIFAR10, Transformers on LM1B). Public codebase can be found in `https://github.com/google/init2winit`.
- **XLA:** Tuning LLM latencies over a variety of different settings for the XLA compiler. Such hyperparameters affect sharding, partitioning, memory consumption, and data limits when placing LLM models on accelerators. Such hyperparameters can be found in OpenXLA `https://github.com/openxla/xla`.
- **L2DA:** Hardware-specific hyperparameters such as memory depth and input-output bandwidths, used to optimize computer hardware performance or reduce costs. Many such similar hyperparameters can be found in `https://github.com/srivatsankrishnan/oss-arch-gym`.

### C.2   String Representations

Table 4 contains example string representations of $x$ for different regression task families.

| Task Family | Example Representations |
|---|---|
| BBOB | `[0.3221, -4.2113, 3.1212, 1.5621]` |
| AutoML | `batch_size:128, ml_feature_selection_threshold:0.05, model_type:'DNN_ESTIMATOR',` `activation_fn:'selu', batch_norm:'False', bucketization_strategy:'mdl',` `dropout:0.071, hidden_units:359` |
| Init2Winit | `lr_hparams.base_lr:0.0696, opt_hparams.0.hps.one_minus_b1:0.2823,` `opt_hparams.0.hps.one_minus_b2:0.0432, opt_hparams.1.hps.weight_decay:0.0023` |
| XLA | `auto_cross_replica_sharding:'False', rematerialization_percent_shared_memory_limit:97,` `spmd_threshold_for_windowed_einsum_mib:100000, ...` |
| L2DA | `input_activation_memory_depth:11.0, instruction_memory_depth:15.0,` `io_bandwidth_gbps:4.321, narrow_memory_capacity_bytes:21.0, ...` |

Table 4: Example $x$ representations from each of the regression task families. '...' denotes that there are actually more parameters, but we omit them due to length.

