# OpenReview forum: "Understanding LLM Embeddings for Regression"
_TMLR — Accepted by TMLR_

### Review · Reviewer_YbPJ · 2024-12-16

**Summary Of Contributions:**

The paper investigates the use of LLM embeddings in regression tasks.

It provides analysis of LLM embeddings' effectiveness in regression tasks compared to traditional feature engineering approaches, particularly for high-dimensional problems.

Extensive experiments have been performed across both synthetic benchmarks (BBOB functions) and real-world regression tasks.

**Audience:**

Yes

**Claims And Evidence:**

Yes

**Requested Changes:**

Provide more theoretical justification for the dimensionality benefits observed.

Expand analysis of cases where LLM embeddings underperform.

Include more diverse real-world regression tasks beyond specialized domains.

**Strengths And Weaknesses:**

Strengths:

Rigorous empirical methodology with comprehensive experiments across diverse regression tasks;

Novel analytical framework (NLFD) for understanding embedding properties;

Clear and detailed ablation studies examining various factors affecting performance.

Weaknesses:

It would be good for authors to provide some insights on why LLM embeddings work well in certain high-dimensional cases but not others;

The paper could benefit from more theoretical analysis, e.g., the relationship between embedding dimension and regression performance;

The real-world tasks are all from relatively specialized domains; could include more general regression problems;

---

> ### Author Response · Authors · 2025-01-07
> **Follow-up: Thank you!**
>
> Hi Reviewer YbPJ, thanks for your positive review of our paper and suggested comments. Weaknesses addressed below:
>
> * “Why LLM embeddings work well in certain high-dimensional cases but not others”
>     * We believe this is mostly an empirical result dependent on the objective landscape. This would be analogous to “XGBoost performs well on some tasks but not others”. At the end of the day, some objective landscapes are more smooth than others, which we’ve shown in Section 3.2.
> * "Theoretical Analysis"
>     * Good question - in the new Appendix A6, we also empirically found that even _vocabulary embeddings_ themselves are dimensionally robust as features. We can arrive to a statement that $||\phi(x) - \phi(x') || < L ||x - x'||$ where $L$ is a function of the vocabulary embedding table, $\phi$ is the average pool of vocabulary embeddings, and $x$ is a real number, assuming digit-by-digit tokenization.
>     * This implies that average-pool embeddings at least have a maximal distortion rate $L$. But since $L$ is dependent on the vocabulary table and later on the Transformer if extending this statement to a full model forward pass, it gets very difficult to quantify with pure theory, given too many moving parts (attention layers with layer-norms, pretraining, etc.). But this is worth investigating more in future work.
> * "Real-World Regression"
>   * Clarified in Appendix C1. We emphasize that our "real world tasks" in the main body should be considered truly "real world", as they are sourced from open-source packages, with the following links: [AutoML](https://cloud.google.com/vertex-ai?#train-models-with-minimal-ml-expertise), [Init2Winit](https://github.com/google/init2winit), [XLA](https://github.com/openxla/xla), [L2DA](https://github.com/srivatsankrishnan/oss-arch-gym). Regression over experimental evaluations is a very common subject within Bayesian Optimization and industrial engineering in general.
>   * In Appendix C3, we also spent a serious effort on using OpenML regression tasks (OpenML-CTR23 and AMLB) but ultimately these tasks were too diverse and also too few to make a statistically significant statement. Compare these to our "Real World Tasks" in the main body, which have 50-75 tasks per family, with all tasks being homogenous within each family, which allows to use make more definitive conclusions and patterns.
>
> References:
>  * OpenML-CTR23: https://openreview.net/pdf?id=HebAOoMm94
>  * AMLB: https://arxiv.org/pdf/2207.12560

---

### Review · Reviewer_SWar · 2024-12-23

**Summary Of Contributions:**

This paper explores the use of embeddings from LLMs as features for regression tasks, providing a comprehensive analysis of their behavior.

* It demonstrates that LLM embeddings maintain strong regression performance even in high-dimensional data scenarios, where traditional feature representations typically degrade in quality. The authors also illustrate some cases where traditional feature representations are better than LLM based features

* It shows that LLM embeddings over numeric data inherently preserve Lipschitz continuity and smoothness in feature space, facilitating effective regression using downstream models such as MLPs. Additionally, the authors build a bridge between the Lipschitz continuity of the feature representation and the success of regression.

* This paper quantifies the effects of LLM-related factors (model size, pre-training quality, and input formatting) on regression performance, revealing that improvements in language understanding do not always lead to better regression outcomes.

**Audience:**

Yes

**Broader Impact Concerns:**

None.

**Claims And Evidence:**

Yes

**Requested Changes:**

I asked some questions and changes in the strengths and weaknesses section.

**Strengths And Weaknesses:**

Strenghts: I think the organization of the paper is good and the transitions between sections are smooth. The contributions are strong. Particularly, the explanation to the success of the feature representations in the regression task using the Lipschitz continuity of the representation over a dataset in Figure 4 is insightful.

Weaknesses: I have the following suggestions and questions:

* It would be very beneficial to explain the BBOB suite because this dataset is utilized a lot throughout the paper. It is very beneficial for a reader to know the basics of this dataset similar to other dataset AutoML, Init2Winit, etc.

* In Section 2, the authors explain how they achieve a feature in $\mathbb{R}^d$. In the forth step of this explanation, the authors mention pooling. Could the authors explain how they apply the pooling. From the next page, I understand that they utilize average pooling. However, it would be better if they explain it under these steps. I am also curious what would happen when they utilize different pooling. I think this is particularly important when the authors analyze the Lipschitz continuity of the feature space.

* The authors inform that the regression is performed using a MLP that is similar for both $\Phi_{\text{LLM}}$ and $\Phi_{\text{trad}}$. However, the authors also compare the LLM-based features with XGBoost as well. I am curious what would happen when they utilize different kind of regression such as XGBoost instead of MLP. In some cases, XGBoost outpetforms MLP. That is why I am curious about the usage of XGBoost instead of MLP. Could the authors explain these behavior in such cases?

* Do the authors have any explanation of Figure 7? Specifically, why does the random initialized model is pretty succesful in AutoML and L2DA? I think this part needs more explanation.

---

> ### Author Response · Authors · 2025-01-07
> **Follow-up: Thank you!**
>
> Hi Reviewer SWar, thanks a lot for your positive feedback and support of our claims and evidence. Weaknesses addressed concisely below:
> * Explain the BBOB suite:
>     * Done, in Appendix C.1 and referenced in main body.
> * Explaining Pooling:
>     * Done, in Appendix B2 and referenced in main body.
> * Additional Pooling (e.g. MaxPooling) Comparison Experiments
>     * Done, in Appendix A4 with (Average-Pooling, Max-Pooling, Last Token). Max-Pooling performs slightly worse than Average-Pooling but is still dimensionally robust. Last-Token embedding performs the absolute worst and is _not_ dimensionally robust.
> * XGBoost on LLM embedding Experiments
>     * Done, in Appendix A5. Using LLM embeddings as features, XGBoost performs worse but is quite correlated with MLP results. XGBoost also remains dimensionally robust. This implies the quality of LLM embeddings is a fundamental property, regardless of the regression head.
> * Randomly initialized model being successful needs explanation:
>     * We believe that this is part of our contributions actually - we empirically show that in certain cases, language pretraining indeed _doesn't_ help for regression. In the new Appendix A6 where we applied a similar analysis for BBOBs, we also find that sometimes performing a full model forward pass doesn't help over simply using vocabulary embeddings either. Since LLMs have many moving parts, it's not possible to theoretically explain why a specific regression task performs well or worse unfortunately.

---

> > ### Comment · Reviewer_SWar · 2025-01-29
> >
> > I want to thank the authors for their clarifications to my comments. The authors satisfactorily answered all of my questions.

---

### Review · Reviewer_rvsG · 2024-12-30

**Summary Of Contributions:**

The paper proposes an LLM-based regression method that utilizes embeddings rather than decoding approaches. The authors evaluate their method on various black-box optimization benchmark problems and some real-world datasets, comparing with non-LLM regression baselines (XGBoost and MLP).

**Audience:**

Yes

**Claims And Evidence:**

No

**Requested Changes:**

- Evaluate on high-dimensional (≈100D) real-world benchmarks where the method shows promise

- Add comparisons with LLM-based regression methods (Song et al., 2024; Vacareanu et al., 2024)

- Present Figure 7 results on BBOB tasks across different DOFs

- Compare data size robustness (Figure 9) with non-LLM baselines

**Strengths And Weaknesses:**

## Strengths:
- The authors present a novel perspective by attempting to use LLM embeddings for regression tasks, which is an innovative idea.

- The robustness of the proposed method to the number of data points and higher dimensional problems, as seen in Figure 9 and Figure 2, are interesting and promising observations for further exploration.


## Weaknesses:

**Major Concerns:**
- The paper's motivation for using LLMs in regression is unclear. While LLMs operate on discrete tokens, regression deals with continuous spectral properties. LLMs' domain knowledge as additional prior may be one potential benefit. However, the current reported results contradict these. For example, random LLM initialization performs well (as in Figure 7), and feature names provide no meaningful advantage (as in Figure 8). This raises fundamental questions about the necessity of LLMs for this task, as neither pre-training knowledge nor domain knowledge appear to contribute significantly to performance.

- Table 1 shows that the LLM-based method underperforms against the non-LLM baseline on real-world problems, with success rates below 50% across all datasets and mostly under 20%. Given its success on high-dimensional black-box problems (in Figure 2), testing on real-world benchmarks with similar dimensionality (≈100D) would be more appropriate.

- Despite the motivations over LLM decoding-based regression approaches, no comparisons are made with such methods. The evaluation needs comparisons with recent LLM-based regression approaches (Song et al., 2024; Vacareanu et al., 2024) to validate the contributions. Current evaluation only includes non-LLM baselines (XGBoost, MLP).

- Also, the selected real-world benchmarks in this paper are not that common in regression analysis. It would be helpful to provide evaluation on more common regression benchmarks such as OpenMLE or PMLB.


**Minor Concerns:**
- I would suggest presenting Figure 7 results also on BBOB tasks across different DOFs to clarify the role of pre-training, especially for higher-dimensional problems where the proposed method outperforms non-LLM baselines as shown in Figure 2.
- Figure 9 is showing the proposed method's robustness to data size. Comparing this with non-LLM baselines could also highlight a potential advantage of LLMs.
- On page 14, Section A.2, Figure 11 is mistakenly referenced as Table 11.

---

> ### Author Response · Authors · 2025-01-07
> **Follow-up: Many experiments conducted**
>
> Hi Reviewer rvsG, thanks for providing your suggestions. We’ve addressed all of them (experimentally or with paper rewriting) below:
>
> **Major Concerns**
> * "Necessity of LLMs for this task"
>     * Clarification: Our paper is an investigation paper, not a methods paper. We investigate the promising behaviors of LLM embeddings for regression, but currently don’t actually advocate for using them on all regression problems (mentioned in Section 4 limitations).
> * “Discrete tokens” + “Neither pre-training nor domain knowledge appear to contribute”
>     * Agreed we’d _expect_ tokenization to have worse inductive biases than continuous features for regression. But as it turns out, Sections 3.1 and 3.2 show that this discretization still has continuity properties, sometimes _better_ than continuous features. Showing this phenomenon is a core contribution of our paper.
>     * Note also that tokenization avoids the need for feature engineering, which in itself can be a huge future benefit regardless of domain knowledge.
> * High-Dimensional Open-Source regression tasks:
>   * Added to Appendix C3. We spent a serious effort on using OpenML regression tasks (OpenML-CTR23 and AMLB) but ultimately these tasks were too diverse and also too few to make a statistically significant statement. Compare these to our "Real World Tasks" in the main body, which have 50-75 tasks per family, with all tasks being homogenous within each family, which allows to use make more definitive conclusions and patterns.
>   * Also clarified in Appendix C1. Please note that our real world tasks are also sourced from open-source packages, with the following links: [AutoML](https://cloud.google.com/vertex-ai?#train-models-with-minimal-ml-expertise), [Init2Winit](https://github.com/google/init2winit), [XLA](https://github.com/openxla/xla), [L2DA](https://github.com/srivatsankrishnan/oss-arch-gym). Regression over experimental evaluations is a very common subject within Bayesian Optimization and industrial engineering in general.
> * “Comparisons with recent LLM-based regression approaches”
>     * We reiterate that our paper is purely an investigation paper around embeddings and not a competition with all LLM-based regression methods. Adding comparisons to (Song et al. 2024, Vacareanu et al. 2024) would require a huge amount of experiments and won’t affect our core message around embedding continuity.
>
> **Minor Concerns**
> * Figure 7 results on BBOBs
>   * Done, added to Appendix A6 and referred to in the main body. The outcomes are similar to Figure 7 (full forward pass by pretrained model beats random-init and vocabulary embeddings). An interesting observation is that random-init and vocabulary embeddings are _also dimensionally robust_.
> * Figure 9 comparison to non-LLM baselines
>     * Clarification: The Y-axis is the gap between using T5-XXL features vs traditional features, on top of a MLP head, and MLP with traditional features is a non-LLM baseline.
> * Typos: Fixed, thanks!
>
> References:
>  * OpenML-CTR23: https://openreview.net/pdf?id=HebAOoMm94
>  * AMLB: https://arxiv.org/pdf/2207.12560
>
> Please let us know if you have any other concerns, and if our response and edits have changed your rating of our paper.

---

> > ### Comment · Reviewer_rvsG · 2025-01-14
> > **Response from Reviewer rvsG**
> >
> > I want to thank the authors for clarification on some points and additional experiments.  They strengthen and clarify the work's contributions.
> >
> > > Figure 7 results on BBOBs. The outcomes are similar to Figure 7 (full forward pass by pretrained model beats random-init and vocabulary embeddings). An interesting observation is that random-init and vocabulary embeddings are also dimensionally robust.
> > >
> > The results in Appendix A6 are interesting. I suggest including this in the main body, as it highlights the more significant and consistent gap between pre-trained embeddings and random-init or vocabulary embeddings, especially for lower DOFs (e.g., 5). This aligns with your findings on the robustness of LLMs to high-dimensional data and raises interesting future directions regarding pre-training's role across varying regression complexities.
> >
> > > We spent a serious effort on using OpenML regression tasks (OpenML-CTR23 and AMLB) but ultimately these tasks were too diverse and also too few to make a statistically significant statement. Compare these to our "Real World Tasks" in the main body, which have 50-75 tasks per family, with all tasks being homogenous within each family, which allows to use make more definitive conclusions and patterns.
> > >
> > This makes sense. Thanks for the clarification.

---

### Author Response · Authors · 2025-01-07
**General Response**

Hi Reviewers, thank you all for your informative suggestions and feedback. To summarize the changes in the paper:

**Additional Experiments**
* Appendix A4 contains ablations between using (Average Pooling, Max Pooling, Last Token logits) - addressing Reviewer SWar
* Appendix A5 contains ablations on using XGBoost over LLM embeddings too - addressing Reviewer SWar
* Appendix A6 contains comparisons to random-init and vocab embedding results applied to BBOB tasks - addressing Reviewer rvsG
* Appendix C3 contains results on OpenML regression tasks, showing that their datasets unfortunately are not comprehensive enough to make any definitive conclusions - addressing Reviewers rvsG and YbPJ

**Additional Writing**
* Appendix B2 contains pooling descriptions - addressing Reviewer SWar
* Appendix C1 contains extended descriptions of our regression tasks and their open-sourcing - addressing all reviewers

We believe these additions and clarifications significantly strengthen the paper and directly address the reviewers' valuable feedback, while also acknowledging limitations and opening doors for future research. Thanks!

---

### Decision · Action_Editor_Wq9Y · 2025-02-12

**Recommendation:** Accept as is

**Comment:**

This paper investigates the use of LLM embeddings as features for high-dimensional regression, comparing with baselines such as XGBoost and MLP on black-box optimization benchmark problems and real-world datasets. The authors present cases in which LLM embeddings outperform traditional feature representations and also cases in which the traditional features perform better. Remarkably, the LLM embeddings are shown to preserve Lipschitz continuity in feature space. Furthermore, it is shown that an improvement in the LLM (in terms of model size and language modeling) does not necessarily lead to better regression performance.

During the revision process, the authors have successfully addressed the concerns of the reviewers, also providing a number of additional experiments. All the three reviewers have expressed a positive recommendation. I concur and recommend acceptance.

**Audience:**

The paper is relevant and timely, and it will be interesting for TMLR's audience.

**Claims And Evidence:**

The empirical findings provided by the authors are comprehensive, and they clearly provide enough evidence to justify their claims.